# Nonlinear Fokker–Planck Equations, H-Theorem and Generalized Entropy of a Composed System

**DOI:** 10.3390/e25091357

**Published:** 2023-09-20

**Authors:** Luiz R. Evangelista, Ervin K. Lenzi

**Affiliations:** 1Dipartimento di Scienza Applicata e Tecnologia (DISAT) del Politecnico di Torino, Corso Duca degli Abruzzi 24, 10129 Torino, Italy; 2Istituto dei Sistemi Complessi del Consiglio Nazionale delle Ricerche (ISC-CNR) co Politecnico di Torino, Corso Duca degli Abruzzi 24, 10129 Torino, Italy; 3Departamento de Física, Universidade Estadual de Maringá, Avenida Colombo, 5790, Maringá 87020-900, Paraná, Brazil; 4Departamento de Física, Universidade Estadual de Ponta Grossa, Avenida General Carlos Cavalcanti, 4748, Ponta Grossa 84030-900, Paraná, Brazil; eklenzi@uepg.br

**Keywords:** generalized entropy, H–theorem, entropy production, nonlinear Fokker–Planck equation

## Abstract

We investigate the dynamics of a system composed of two different subsystems when subjected to different nonlinear Fokker–Planck equations by considering the H–theorem. We use the H–theorem to obtain the conditions required to establish a suitable dependence for the system’s interaction that agrees with the thermodynamics law when the nonlinearity in these equations is the same. In this framework, we also consider different dynamical aspects of each subsystem and investigate a possible expression for the entropy of the composite system.

## 1. Introduction

Thermodynamics and statistical mechanics have entropy as a fundamental tool connecting the properties of a system from the particles’ microscopic dynamics with macroscopic quantities and, consequently, with thermodynamic quantities. The concept of entropy started with Clausius’s studies of thermal machines [1]. Subsequently, the Boltzmann and Gibbs works incorporated the concept of probability, building up the fundamentals of statistical mechanics [2,3,4]. It has been successfully applied in many contexts, where the fundamental basis is the molecular chaos hypothesis, which assumes the close-range interaction of molecules and the absence of memory in the collision of particles [5,6]. However, for many physical systems (e.g., fractal and self-organizing structures), conditions for the fulfillment of the molecular chaos hypothesis are not observed as well as the range of the interactions, which are long-ranged [7,8,9]. These points have motivated the analysis of extensions for thermodynamics and statistical physics to cover these scenarios. As an example, Tsallis has proposed an extension of the entropy [10], which has been systematically applied in many contexts such as black holes [11], the electrocaloric effect in quantum dots [12], chemotaxis of biological populations [13], Bose–Einstein condensation [14,15], and stimulated the analysis of other entropies [16,17,18,19,20]. More applications can be found in Refs. [21,22,23,24,25,26]. These entropies verify the H–theorem [27,28,29,30,31], which represents an important result of nonequilibrium statistical mechanics by ensuring that a system will reach an equilibrium after a long time evolution. The H–theorem establishes a connection between the dynamics and entropy, which may be used to investigate the dynamics behind the law of additivity for the different entropies. In this framework, by considering a nonlinear Fokker–Planck equation, the H–theorem can show how the entropy additivity laws can be obtained when a system composed of many subsystems is taken into account. In addition, it can also allow us to obtain the equilibrium distributions.

Here, we investigate through the H–theorem the conditions on the dynamics equations, i.e., nonlinear Fokker–Planck equations [32,33,34,35], for each subsystem of a composed system to reach the equilibrium condition. The results show that generalized entropies imply a coupling between the nonlinear equations. The distributions that emerge from these dynamics equations have a power-law behavior, where each subsystem modifies the other. We also investigate the entropy production for this system. These developments are presented in Section 2. In Section 3, we present our discussions and conclusions.

## 2. The Problem

Let us start our analysis by establishing the nonlinear Fokker–Planck equations connected to the dynamics of each subsystem of a composed system. They are
(1)$$\begin{array}{c} \frac{\partial }{\partial t}\rho_{1}\left(x_{1},t\right)=\Gamma \frac{\partial^{2}}{\partial x_{1}^{2}}P_{1}\left(\rho_{1},t\right)-\frac{\partial }{\partial x_{1}}\left[F_{1}\left(x_{1}\right)\rho_{1}\left(x_{1},t\right)\right] \end{array}$$
and
(2)$$\begin{array}{c} \frac{\partial }{\partial t}\rho_{2}\left(x_{2},t\right)=\Gamma \frac{\partial^{2}}{\partial x_{2}^{2}}P_{2}\left(\rho_{2},t\right)-\frac{\partial }{\partial x_{2}}\left[F_{2}\left(x_{2}\right)\rho_{2}\left(x_{2},t\right)\right], \end{array}$$
where $F_{i}\left(x_{i}\right)$, with i=1 or 2, represents the external force, i.e., $F_{i}=-\partial_{x_{i}}\phi_{i}\left(x_{i}\right)$ and $\phi_{i}\left(x_{i}\right)$ is a potential energy, while Γ stands for a generic diffusion coefficient. Notice that $P_{1}\left(\rho_{1},t\right)$ and $P_{2}\left(\rho_{2},t\right)$ present in the diffusive term may have the same form or a different form. Particular choices of $P_{i}\left(\rho_{i},t\right)$ have been successfully analyzed in several problems such as in porous media [36], anomalous diffusion [37], overdamped systems [38], and the Boltzmann equation endowed with a correlation term [39]. In Equations (Equation 1) and (Equation 2), $P_{i}\left(\rho_{i},t\right)$ will be determined by the H–theorem in connection with the entropic form used to describe the combination of subsystems 1 and 2. It is worth pointing out that the different possibilities may be considered by allowing us to obtain different results for the composite system of 1 and 2 subsystems, as discussed in Refs. [28,29]. However, the combination of these equations, which represent the subsystem 1 and 2, in connection with thermostatistics (e.g., the nonextensive statistics [40]) requires careful analysis with direct consequences on the entropic additivity and zeroth law [41,42,43]. To accomplish this task, we consider general scenarios with different dynamics to investigate possible conditions to Equations (Equation 1) and (Equation 2) to allow a thermostatistics context.

### 2.1. H-Theorem

We start our analysis in terms of the H–theorem first by considering $P_{1}\left(\rho_{1}\right)$ and $P_{2}\left(\rho_{2}\right)$ with the same functional form. Afterwards, we consider $P_{1}\left(\rho_{1}\right)$ and $P_{2}\left(\rho_{2}\right)$ with a different functional form. Each one of these cases has different implications for the entropy related to the composed system formed by the systems 1 and 2, with the dynamics given in terms of Equations (Equation 1) and (Equation 2). Following Refs. [28,29,31], we analyze the behavior of the time derivative of the Helmholtz free energy. This free energy is defined by F=U−TS, with the internal energy, *U*, given by
(3)$$\begin{array}{c} U=\int_{-\infty }^{\infty }dx_{1}\int_{-\infty }^{\infty }dx_{2}\left[\phi_{1}\left(x_{1}\right)+\phi_{2}\left(x_{2}\right)\right]\rho_{1}\left(x_{1},t\right)\rho_{2}\left(x_{2},t\right) \end{array}$$
and the entropy, *S*, expressed in terms of an arbitrary function
(4)$$\begin{array}{c} S=k\int_{-\infty }^{\infty }dx_{1}\int_{-\infty }^{\infty }dx_{2}s\left(\rho_{1},\rho_{2}\right). \end{array}$$
Note that Equations (Equation 3) and (Equation 4) represent the total internal energy and the entropy of the system composed of two subsystems governed by Equations (Equation 1) and (Equation 2), respectively.

By using the previous equations, the total free energy of the system is given by
(5)$$\begin{array}{c} F=\int_{-\infty }^{\infty }\int_{-\infty }^{\infty }dx_{1}dx_{2}\left[\Psi \left(x_{1},x_{2}\right)\rho_{1}\left(x_{1},t\right)\rho_{2}\left(x_{2},t\right)-kTs\left(\rho_{1},\rho_{2}\right)\right], \end{array}$$
with $\Psi \left(x_{1},x_{2}\right)=\phi_{1}\left(x_{1}\right)+\phi_{2}\left(x_{2}\right)$. Before determining the time derivative of Equation (Equation 5), we assume that $P_{1}\left(\rho_{1},t\right)$ and $P_{2}\left(\rho_{2},t\right)$ have essentially the same functional forms and the entropy is a function of the product of the probability densities related to each subsystem, i.e., $s\left(\rho_{1},\rho_{2}\right)=s\left(\rho_{1}\rho_{2}\right)$. It is then possible to show that
(6)$$\begin{array}{c} \frac{d}{dt}F=\int_{-\infty }^{\infty }\int_{-\infty }^{\infty }dx_{1}dx_{2}\left[\Psi \left(x_{1},x_{2}\right)-kT\frac{\partial }{\partial \rho_{12}}s\left(\rho_{12}\right)\right]\frac{\partial }{\partial t}\left[\rho_{1}\left(x_{1},t\right)\rho_{2}\left(x_{2},t\right)\right], \end{array}$$
where $\rho_{12}=\rho_{1}\rho_{2}$, and
(7)$$\begin{array}{ccc} \frac{d}{dt}F & = & \int_{-\infty }^{\infty }\int_{-\infty }^{\infty }dx_{1}dx_{2}\left\{\Psi \left(x_{1},x_{2}\right)\rho_{2}-kT\rho_{2}\frac{\partial }{\partial \rho_{12}}s\left(\rho_{12}\right)\right\}\\ & \times & \frac{\partial }{\partial x_{1}}\left\{\Gamma \frac{\partial }{\partial x_{1}}P_{1}\left(\rho_{1},t\right)-F_{1}\left(x_{1}\right)\rho_{1}\left(x_{1},t\right)\right\}\\ & + & \int_{-\infty }^{\infty }\int_{-\infty }^{\infty }dx_{1}dx_{2}\left\{\Psi \left(x_{1},x_{2}\right)\rho_{1}-kT\rho_{1}\frac{\partial }{\partial \rho_{12}}s\left(\rho_{12}\right)\right\}\\ & \times & \frac{\partial }{\partial x_{2}}\left\{\Gamma \frac{\partial }{\partial x_{2}}P_{2}\left(\rho_{2},t\right)-F_{2}\left(x_{2}\right)\rho_{2}\left(x_{2},t\right)\right\}. \end{array}$$
After integration by parts and applying the conditions $\rho_{i}\left(x\to \pm \infty ,t\right)\to 0$ and $\partial_{x}\rho_{i}\left(x\to \pm \infty ,t\right)\to 0$, we obtain
(8)$$\begin{array}{ccc} \frac{d}{dt}F & = & -\int_{-\infty }^{\infty }\int_{-\infty }^{\infty }dx_{1}dx_{2}\left\{\frac{\partial }{\partial x_{1}}\phi_{1}\left(x_{1}\right)\rho_{2}-kT\rho_{2}^{2}\frac{\partial \rho_{1}}{\partial x_{1}}\frac{\partial^{2}}{\partial \rho_{12}^{2}}s\left(\rho_{12}\right)\right\}\\ & \times & \left\{\Gamma \frac{\partial }{\partial x_{1}}P_{1}\left(\rho_{1},t\right)-F_{1}\left(x_{1}\right)\rho_{1}\left(x_{1},t\right)\right\}\\ & - & \int_{-\infty }^{\infty }\int_{-\infty }^{\infty }dx_{1}dx_{2}\left\{\frac{\partial }{\partial x_{2}}\phi_{2}\left(x_{2}\right)\rho_{1}-kT\rho_{1}^{2}\frac{\partial \rho_{2}}{\partial x_{2}}\frac{\partial^{2}}{\partial \rho_{12}^{2}}s\left(\rho_{12}\right)\right\}\\ & \times & \left\{\Gamma \frac{\partial }{\partial x_{2}}P_{2}\left(\rho_{2},t\right)-F_{2}\left(x_{2}\right)\rho_{2}\left(x_{2},t\right)\right\}. \end{array}$$

Now, let us focus on the term
(9)$$\begin{array}{c} \Gamma \frac{\partial }{\partial x_{i}}P_{i}\left(\rho_{i},t\right)-F_{i}\left(x_{i}\right)\rho_{i}\left(x_{i},t\right), \end{array}$$
where i=1,2, which will be directly connected with the properties of the entropy of the composite system. To proceed, we consider that
(10)$$\begin{array}{c} P_{i}\left(\rho_{i},t\right)=D_{j,\gamma }\left(t\right)\rho_{i}^{\gamma }\left(x_{i},t\right)+D_{j,\nu }\left(t\right)\rho_{i}^{\nu }\left(x_{i},t\right), \end{array}$$
with j≠i, j=1,2, and
(11)$$\begin{array}{c} D_{j,\gamma }\left(t\right)=\alpha_{\gamma }\int_{-\infty }^{\infty }dx_{j}\rho_{j}^{\gamma }\left(x_{j},t\right)\;\mathrm{and}\;D_{j,\nu }\left(t\right)=\alpha_{\nu }\int_{-\infty }^{\infty }dx_{j}\rho_{j}^{\nu }\left(x_{j},t\right), \end{array}$$
to be able to cover different scenarios, where $\alpha_{\gamma }$ and $\alpha_{\nu }$ are constants. Note that the choice of the $D_{j,\gamma }\left(t\right)$ and $D_{j,\nu }\left(t\right)$ implies that each subsystem influences the other. This aspect of the problem can be associated to the feature that the nonlinearity present in Equations (Equation 1) and (Equation 2) introduces additional interactions between the subsystems during the thermalization process, where each subsystem works as an additional thermal bath to the other. By using the previous equations, we have
(12)$$\begin{array}{ccc} \frac{d}{dt}F & = & -\int_{-\infty }^{\infty }dx_{1}\frac{1}{\rho_{1}}\left\{\int_{-\infty }^{\infty }dx_{2}\left[\frac{\partial }{\partial x_{1}}\phi_{1}\left(x_{1}\right)\rho_{2}\rho_{1}-kT\rho_{2}^{2}\rho_{1}\frac{\partial \rho_{1}}{\partial x_{1}}\frac{\partial^{2}}{\partial \rho_{12}^{2}}s\left(\rho_{12}\right)\right]\right.\\ & \times & \left.\int_{-\infty }^{\infty }dx_{2}\left[\frac{\partial }{\partial x_{1}}\phi_{1}\left(x_{1}\right)\rho_{1}\rho_{2}+\Gamma \frac{\partial \rho_{1}}{\partial x_{1}}\rho_{2}\frac{\partial }{\partial \rho_{12}}\left(\alpha_{\gamma }\rho_{2}^{\gamma }\rho_{1}^{\gamma }+\alpha_{\nu }\rho_{2}^{\nu }\rho_{1}^{\nu }\right)\right]\right\}\\ & - & \int_{-\infty }^{\infty }dx_{2}\frac{1}{\rho_{2}}\left\{\int_{-\infty }^{\infty }dx_{1}\left[\frac{\partial }{\partial x_{2}}\phi_{2}\left(x_{2}\right)\rho_{1}\rho_{2}-kT\rho_{1}^{2}\rho_{2}\frac{\partial \rho_{2}}{\partial x_{2}}\frac{\partial^{2}}{\partial \rho_{12}^{2}}s\left(\rho_{12}\right)\right]\right.\\ & \times & \int_{-\infty }^{\infty }dx_{1}\left.\left[\frac{\partial }{\partial x_{2}}\phi_{2}\left(x_{2}\right)\rho_{1}\rho_{2}+\Gamma \frac{\partial \rho_{2}}{\partial x_{2}}\rho_{1}\frac{\partial }{\partial \rho_{12}}\left(\alpha_{\gamma }\rho_{2}^{\gamma }\rho_{1}^{\gamma }+\alpha_{\nu }\rho_{2}^{\nu }\rho_{1}^{\nu }\right)\right]\right\}. \end{array}$$
We verify that
(13)$$\begin{array}{ccc} \frac{d}{dt}F\leq 0\;\;\;\mathrm{for}\;\;\;-kT\rho_{12}\frac{\partial^{2}}{\partial \rho_{12}^{2}}s\left(\rho_{12}\right) & = & \Gamma \frac{\partial }{\partial \rho_{12}}\left(\alpha_{\gamma }\rho_{2}^{\gamma }\rho_{1}^{\gamma }+\alpha_{\nu }\rho_{2}^{\nu }\rho_{1}^{\nu }\right)\\ & = & \Gamma \frac{\partial }{\partial \rho_{12}}\left(\alpha_{\gamma }\rho_{12}^{\gamma }+\alpha_{\nu }\rho_{12}^{\nu }\right), \end{array}$$
which implies
(14)$$\begin{array}{ccc} \frac{d}{dt}F & = & -\int_{-\infty }^{\infty }dx_{1}\frac{1}{\rho_{1}}{\left\{\int_{-\infty }^{\infty }dx_{2}\left[\frac{\partial }{\partial x_{1}}\phi_{1}\left(x_{1}\right)\rho_{2}\rho_{1}-kT\rho_{2}^{2}\rho_{1}\frac{\partial \rho_{1}}{\partial x_{1}}\frac{\partial^{2}}{\partial \rho_{12}^{2}}s\left(\rho_{12}\right)\right]\right\}}^{2}\\ & - & \int_{-\infty }^{\infty }dx_{2}\frac{1}{\rho_{2}}{\left\{\int_{-\infty }^{\infty }dx_{1}\left[\frac{\partial }{\partial x_{2}}\phi_{2}\left(x_{2}\right)\rho_{1}\rho_{2}-kT\rho_{1}^{2}\rho_{2}\frac{\partial \rho_{2}}{\partial x_{2}}\frac{\partial^{2}}{\partial \rho_{12}^{2}}s\left(\rho_{12}\right)\right]\right\}}^{2}. \end{array}$$
Consequently, by solving Equation (Equation 13) with Γ=kT under the conditions defined in Refs. [28,29,30,31], we obtain
(15)$$\begin{array}{c} s\left(\rho_{12}\right)=\frac{\alpha_{\gamma }}{\gamma -1}\left(\rho_{12}-\rho_{12}^{\gamma }\right)+\frac{\alpha_{\nu }}{\nu -1}\left(\rho_{12}-\rho_{12}^{\nu }\right). \end{array}$$
The entropy for the composite system is given by
(16)$$\begin{array}{ccc} S & = & \frac{\alpha_{\gamma }k}{\gamma -1}\int_{-\infty }^{\infty }dx_{1}\int_{-\infty }^{\infty }dx_{2}\left(\rho_{12}-\rho_{12}^{\gamma }\right)\\ & + & \frac{\alpha_{\nu }k}{\nu -1}\int_{-\infty }^{\infty }dx_{1}\int_{-\infty }^{\infty }dx_{2}\left(\rho_{12}-\rho_{12}^{\nu }\right), \end{array}$$
which can also be rewritten as
(17)$$\begin{array}{ccc} S & = & \frac{\alpha_{\gamma }k}{\gamma -1}\int_{-\infty }^{\infty }dx_{1}\int_{-\infty }^{\infty }dx_{2}\left[\rho_{1}\rho_{2}-{\left(\rho_{1}\rho_{2}\right)}^{\gamma }\right]\\ & + & \frac{\alpha_{\nu }k}{\nu -1}\int_{-\infty }^{\infty }dx_{1}\int_{-\infty }^{\infty }dx_{2}\left[\rho_{1}\rho_{2}-{\left(\rho_{1}\rho_{2}\right)}^{\nu }\right] \end{array}$$
and, consequently, as
(18)$$\begin{array}{ccc} S & = & \frac{\alpha_{\gamma }k}{\gamma -1}\left[1-\int_{-\infty }^{\infty }dx_{1}\int_{-\infty }^{\infty }dx_{2}{\left(\rho_{1}\rho_{2}\right)}^{\gamma }\right]\\ & + & \frac{\alpha_{\nu }k}{\nu -1}\left[1-\int_{-\infty }^{\infty }dx_{1}\int_{-\infty }^{\infty }dx_{2}{\left(\rho_{1}\rho_{2}\right)}^{\nu }\right]. \end{array}$$
Equation (Equation 18) has several particular cases, such as the Tsallis and Kaniadakis entropies, depending on the values of the parameters $\alpha_{\gamma }$, $\alpha_{\nu }$, γ, and ν. It is noteworthy that this result preserves the additivity in the Penrose sense [3], i.e., $S\left(\rho_{12}\right)=S\left(\rho_{1}\rho_{2}\right)$ required for a system composed of independent subsystems when the standard entropy is employed.

In the previous context, Equations (Equation 1) and (Equation 2) can be written as follows:(19)$$\begin{array}{c} \frac{\partial }{\partial t}\rho_{1}\left(x_{1},t\right)={\bar{D}}_{2,\gamma }\left(t\right)\frac{\partial^{2}}{\partial x_{1}^{2}}\rho_{1}^{\gamma }\left(x_{1},t\right)+{\bar{D}}_{2,\nu }\left(t\right)\frac{\partial }{\partial x_{1}^{2}}\rho_{1}^{\nu }\left(x_{1},t\right)-\frac{\partial }{\partial x_{1}}\left[F_{1}\left(x_{1}\right)\rho_{1}\left(x_{1},t\right)\right] \end{array}$$
and
(20)$$\begin{array}{c} \frac{\partial }{\partial t}\rho_{2}\left(x_{2},t\right)={\bar{D}}_{1,\gamma }\left(t\right)\frac{\partial^{2}}{\partial x_{2}^{2}}\rho_{2}^{\gamma }\left(x_{2},t\right)+{\bar{D}}_{1,\nu }\left(t\right)\frac{\partial^{2}}{\partial x_{2}^{2}}\rho_{2}^{\nu }\left(x_{2},t\right)-\frac{\partial }{\partial x_{2}}\left[F_{2}\left(x_{2}\right)\rho_{2}\left(x_{2},t\right)\right], \end{array}$$
with ${\bar{D}}_{i,\gamma }\left(t\right)=D_{i,\gamma }\left(t\right)\Gamma$ and ${\bar{D}}_{i,\nu }\left(t\right)=D_{i,\nu }\left(t\right)\Gamma$, by evidencing the influence of one of them on the other. In particular, the terms forming the diffusive part can also be connected with anomalous diffusion processes with different diffusion regimes. The stationary solutions obtained from Equations (Equation 19) and (Equation 20) are given by
(21)$$\begin{array}{c} \frac{\gamma }{\gamma -1}{\bar{D}}_{2,\gamma }\rho_{1,st}^{\gamma -1}\left(x_{1}\right)+\frac{\nu }{\nu -1}{\bar{D}}_{2,\nu }\rho_{1,st}^{\nu -1}\left(x_{1}\right)=C_{1}-\phi_{1}\left(x_{1}\right) \end{array}$$
and
(22)$$\begin{array}{c} \frac{\gamma }{\gamma -1}{\bar{D}}_{1,\gamma }\rho_{2,st}^{\gamma -1}\left(x_{2}\right)+\frac{\nu }{\nu -1}{\bar{D}}_{1,\nu }\rho_{2,st}^{\nu -1}\left(x_{2}\right)=C_{2}-\phi_{2}\left(x_{2}\right), \end{array}$$
where $\lim_{t\to \infty }{\bar{D}}_{i,\gamma }\left(t\right)={\bar{D}}_{i,\gamma }=\mathrm{constant}$, $\phi_{i}\left(x\right)$ are potentials with a minimum, and $C_{i}$ are constants. For the Tsallis entropy, by taking, for simplicity, ${\bar{D}}_{i,\nu }=0$, we have
(23)$$\begin{array}{c} \rho_{1,st}\left(x_{1}\right)=\frac{1}{Z_{1}}{\left[1-\left(\gamma -1\right)\frac{Z_{1}^{\gamma -1}}{\gamma {\bar{D}}_{2,\gamma }}\phi_{1}\left(x_{1}\right)\right]}^{\frac{1}{\gamma -1}}=\frac{1}{Z_{1}}\exp_{\gamma }\left[-\frac{Z_{1}^{\gamma -1}}{\gamma {\bar{D}}_{2,\gamma }}\phi_{1}\left(x_{1}\right)\right] \end{array}$$
and
(24)$$\begin{array}{c} \rho_{2,st}\left(x_{2}\right)=\frac{1}{Z_{2}}{\left[1-\left(\gamma -1\right)\frac{Z_{2}^{\gamma -1}}{\gamma {\bar{D}}_{2,\gamma }}\phi_{1}\left(x_{2}\right)\right]}^{\frac{1}{\gamma -1}}=\frac{1}{Z_{2}}\exp_{\gamma }\left[-\frac{Z_{2}^{\gamma -1}}{\gamma {\bar{D}}_{1,\gamma }}\phi_{2}\left(x_{2}\right)\right], \end{array}$$
where $Z_{i}=1/{\left\{\left[\left(\gamma -1\right)/\left(\gamma {\bar{D}}_{i,\gamma }\right)\right]C_{i}\right\}}^{\frac{1}{\gamma -1}}$ is defined by the normalization condition and ${\bar{D}}_{i,\gamma }=kT\int_{-\infty }^{\infty }dx_{i}\rho_{i,st}^{\gamma }\left(x_{i}\right)$. In the preceding equations, $\exp_{q}\left[x\right]$ is the q−exponential function, defined as follows [40]:(25)$$\begin{array}{c} \exp_{q}\left[x\right]\equiv \left\{\begin{array}{cc} {\left(1+\left(q-1\right)x\right)}^{1/(q-1)} & ,\;\;\;\;\;x>1/(1-q),\\ 0 & ,\;\;\;\;\;x<1/(1-q). \end{array}\right. \end{array}$$
The presence of this function in the previous equations enables the identification of either a short- or a long-tailed behavior of the solution, depending on the value of the parameters γ and ν. Indeed, they may have a compact behavior for γ>1 (or ν>1) due to the *cut-off* required by the *q*-exponential to retain the probabilistic interpretation of the distribution. On the other hand, for γ<1 (or ν>1), the solutions may have the asymptotic limit governed by a power-law behavior, which may also be related to a Lévy distribution [44] and, consequently, asymptotically with the solutions of the fractional Fokker–Planck equations [45], which are asymptotically governed by power-laws.

From the stochastic point of view, Equations (Equation 19) and (Equation 20) are connected with the following Langevin equations:(26)$$\begin{array}{c} {\dot{x}}_{1}=F_{1}\left(x_{1}\right)+\sqrt{2\Gamma \Lambda_{1,2}\left(t\right)}\xi_{1}\left(t\right) \end{array}$$
and
(27)$$\begin{array}{c} {\dot{x}}_{2}=F_{2}\left(x_{2}\right)+\sqrt{2\Gamma \Lambda_{2,1}\left(t\right)}\xi_{2}\left(t\right), \end{array}$$
where $\xi_{1}\left(t\right)$ and $\xi_{2}\left(t\right)$ are connected to the stochastic forces and $\Lambda_{i,j}\left(t\right)=D_{j(i),\gamma (\nu )}\left(t\right)$$\rho_{i(j)}^{\gamma (\nu )}\left(x,t\right)$+$D_{j(i),\nu (\gamma )}\left(t\right)$$\rho_{i(j)}^{\nu (\gamma )}\left(x,t\right)$. In particular, we have
(28)$$\begin{array}{c} \langle \xi_{1}\rangle =\langle \xi_{2}\rangle =0\;\;,\;\;\;\langle \xi_{1}\xi_{2}\rangle =\langle \xi_{2}\xi_{1}\rangle =0 \end{array}$$
and
(29)$$\begin{array}{c} \langle \xi_{1}\left(t\right)\xi_{1}\left(t^{\prime }\right)\rangle \propto \delta \left(t-t^{\prime }\right)\;\;,\;\;\;\langle \xi_{2}\left(t\right)\xi_{2}\left(t^{\prime }\right)\rangle \propto \delta \left(t-t^{\prime }\right). \end{array}$$
The walkers related to this problem can be described, for simplicity, in the absence of external forces, in terms of the following equations [46,47]:(30)$$\begin{array}{c} \rho_{1}\left(x_{1},t+\tau \right)=\int_{-\infty }^{\infty }\Theta_{1,2}\left[x_{1}-x_{1}^{\prime },t;\rho \left(x_{1}-x_{1}^{\prime },t\right)\right]\rho_{1}\left(x_{1}-x_{1}^{\prime },t\right)\Phi \left(x_{1}^{\prime }\right)dx_{1}^{\prime } \end{array}$$
and
(31)$$\begin{array}{c} \rho_{2}\left(x_{2},t+\tau \right)=\int_{-\infty }^{\infty }\Theta_{2,1}\left[x_{2}-x_{2}^{\prime },t;\rho \left(x_{2}-x_{2}^{\prime },t\right)\right]\rho_{2}\left(x_{2}-x_{2}^{\prime },t\right)\Phi \left(x_{2}^{\prime }\right)dx_{2}^{\prime }, \end{array}$$
where
(32)$$\begin{array}{c} \Theta_{i,j}\left[x_{i},t;\rho \left(x_{i},t\right)\right]=\alpha_{\gamma }\int_{-\infty }^{\infty }dx_{j}\rho_{j}^{\gamma }\left(x_{j},t\right)\rho_{i}^{\gamma -1}\left(x_{i},t\right)+\alpha_{\nu }\int_{-\infty }^{\infty }dx_{j}\rho_{j}^{\nu }\left(x_{j},t\right)\rho_{i}^{\nu -1}\left(x_{i},t\right). \end{array}$$
These equations, in the limit τ→0 and $x_{i}^{\prime }\to 0$, yield Equations (Equation 1) and (Equation 2) in the absence of external forces, respectively.

Let us now consider a general case, i.e., the one in which the diffusion terms have a different nonlinear dependence on the distributions. This means that the systems have different dynamical aspects governed by the nonlinear dependence on the distribution present in the diffusive term. By using the preceding equations and having in mind Equation (Equation 5), we may write
(33)$$\begin{array}{ccc} \frac{d}{dt}F & = & \int_{-\infty }^{\infty }\int_{-\infty }^{\infty }dx_{1}dx_{2}\left\{\Psi \left(x_{1},x_{2}\right)\frac{\partial }{\partial t}\left[\rho_{1}\left(x_{1},t\right)\rho_{2}\left(x_{2},t\right)\right]\right.\\ & - & \left.kT\left[\frac{\partial }{\partial \rho_{1}}s\left(\rho_{1},\rho_{2}\right)\frac{\partial }{\partial t}\rho_{1}\left(x_{1},t\right)+\frac{\partial }{\partial \rho_{2}}s\left(\rho_{1},\rho_{2}\right)\frac{\partial }{\partial t}\rho_{2}\left(x_{2},t\right)\right]\right\}, \end{array}$$
which implies
(34)$$\begin{array}{ccc} \frac{d}{dt}F & = & \int_{-\infty }^{\infty }\int_{-\infty }^{\infty }dx_{1}dx_{2}\left\{\Psi \left(x_{1},x_{2}\right)\rho_{2}-kT\frac{\partial }{\partial \rho_{1}}s\left(\rho_{1},\rho_{2}\right)\right\}\\ & \times & \frac{\partial }{\partial x_{1}}\left\{\Gamma \frac{\partial }{\partial x_{1}}P_{1}\left(\rho_{1},t\right)-F_{1}\left(x_{1}\right)\rho_{1}\left(x_{1},t\right)\right\}\\ & + & \int_{-\infty }^{\infty }\int_{-\infty }^{\infty }dx_{1}dx_{2}\left\{\Psi \left(x_{1},x_{2}\right)\rho_{1}-kT\frac{\partial }{\partial \rho_{2}}s\left(\rho_{1},\rho_{2}\right)\right\}\\ & \times & \frac{\partial }{\partial x_{2}}\left\{\Gamma \frac{\partial }{\partial x_{2}}P_{2}\left(\rho_{2},t\right)-F_{2}\left(x_{2}\right)\rho_{2}\left(x_{2},t\right)\right\}. \end{array}$$
After some calculations, it is possible to show that
(35)$$\begin{array}{ccc} \frac{d}{dt}F & = & -\int_{-\infty }^{\infty }dx_{1}\frac{1}{\rho_{1}}\int_{-\infty }^{\infty }dx_{2}\left\{\frac{\partial }{\partial x_{1}}\phi \left(x_{1}\right)\rho_{2}\rho_{1}-kT\left[\rho_{1}\frac{\partial^{2}}{\partial \rho_{1}^{2}}s\left(\rho_{1},\rho_{2}\right)\right]\frac{\partial }{\partial x_{1}}\rho_{1}\right\}\\ & \times & \left\{\left[\Gamma \frac{\partial }{\partial \rho_{1}}P_{1}\left(\rho_{1},t\right)\right]\frac{\partial }{\partial x_{1}}\rho_{1}-F_{1}\left(x_{1}\right)\rho_{1}\right\}\\ & - & \int_{-\infty }^{\infty }dx_{1}\int_{-\infty }^{\infty }dx_{2}\frac{1}{\rho_{2}}\left\{\frac{\partial }{\partial x_{2}}\phi \left(x_{2}\right)\rho_{1}\rho_{2}-kT\left[\rho_{2}\frac{\partial^{2}}{\partial \rho_{2}^{2}}s\left(\rho_{1},\rho_{2}\right)\right]\frac{\partial }{\partial x_{2}}\rho_{2}\right\}\\ & \times & \frac{\partial }{\partial x_{2}}\left\{\left[\Gamma \frac{\partial }{\partial \rho_{2}}P_{2}\left(\rho_{2},t\right)\right]\frac{\partial }{\partial x_{2}}\rho_{2}-F_{2}\left(x_{2}\right)\rho_{2}\right\}. \end{array}$$
Let us analyze in particular the previous equation, for example, for the case
(36)$$\begin{array}{c} P_{1}\left(\rho_{1},t\right)=D_{2,\nu }\left(t\right)\rho_{1}^{\gamma }\left(x_{1},t\right)\;\;\;\mathrm{and}\;\;\;P_{2}\left(\rho_{2},t\right)=D_{1,\gamma }\left(t\right)\rho_{2}^{\nu }\left(x_{2},t\right), \end{array}$$
with
(37)$$\begin{array}{c} D_{2,\nu }\left(t\right)=\frac{1}{\nu -1}\int_{-\infty }^{\infty }dx_{2}\rho_{2}^{\nu }\left(x_{2},t\right)\;\;\;\mathrm{and}\;\;\;D_{1,\gamma }\left(t\right)=\frac{1}{\gamma -1}\int_{-\infty }^{\infty }dx_{1}\rho_{1}^{\gamma }\left(x_{1},t\right), \end{array}$$
which implies different dynamics for each subsystem. We notice that it is possible to take into account different aspects of the dynamics of each subsystem, and every choice has different implications for the total entropy of the composite system. Similar nonlinear Fokker–Planck equations were considered in Ref. [48] from the point of view of analyzing the interaction between the two subsystems. From Equation (Equation 37), we deduce that the entropy needs to satisfy the following equations:(38)$$\begin{array}{c} -\rho_{1}\frac{\partial^{2}}{\partial \rho_{1}^{2}}s\left(\rho_{1},\rho_{2}\right)=\frac{\gamma }{\nu -1}\rho_{2}^{\nu }\rho_{1}^{\gamma -1}\;\;\;\mathrm{and}\;\;\;-\rho_{2}\frac{\partial^{2}}{\partial \rho_{2}^{2}}s\left(\rho_{1},\rho_{2}\right)=\frac{\nu }{\gamma -1}\rho_{2}^{\nu -1}\rho_{1}^{\gamma } \end{array}$$
in order to verify
(39)$$\begin{array}{c} \frac{d}{dt}F\leq 0, \end{array}$$
and, consequently, to satisfy the H–theorem. A solution for the previous system of equations is
(40)$$\begin{array}{c} s\left(\rho_{1},\rho_{2}\right)=\frac{1}{\left(\nu -1)(\gamma -1\right)}\left(\rho_{1}\rho_{2}-\rho_{2}^{\nu }\rho_{1}^{\gamma }\right). \end{array}$$
This result allows us to write the total entropy of this system as follows:(41)$$\begin{array}{c} S=\frac{k}{\left(\nu -1)(\gamma -1\right)}\left[1-\int_{-\infty }^{\infty }dx_{2}\rho_{2}^{\nu }\left(x_{2},t\right)\int_{-\infty }^{\infty }dx_{1}\rho_{1}^{\gamma }\left(x_{1},t\right)\right]. \end{array}$$
It is remarkable that this result for the entropy differs from the preceding one given by Equation (Equation 18), obtained from a different choice of nonlinear Fokker–Planck equations. Equation (Equation 41) results from a combination of different subsystems with different dynamics, which individually have different entropies associated with them. One of the consequences is that the entropy of the composite system, for this specific case, can not be written as $S(\rho_{1}\rho_{2})$, only when γ=ν. Another remarkable point is the connection of Equation (Equation 41) with the composition of Tsallis entropies of different *q*-indices [49,50]. The solution can be found in this framework using the *q*-exponential functions. In particular, it is possible to show that the solution for each nonlinear Fokker–Planck equation, in the absence of external force, is
(42)$$\begin{array}{c} \rho_{1}\left(x_{1},t\right)=\exp_{\gamma }\left[-\beta_{1}\left(t\right)x_{1}^{2}\right]/Z_{1}\left(t\right) \end{array}$$
and
(43)$$\begin{array}{c} \rho_{2}\left(x_{1},t\right)=\exp_{\nu }\left[-\beta_{2}\left(t\right)x_{2}^{2}\right]/Z_{2}\left(t\right), \end{array}$$
with $\beta_{1}\left(t\right)$, $\beta_{2}\left(t\right)$, $Z_{1}\left(t\right)$, and $Z_{2}\left(t\right)$ obtained from the following set of equations:(44)$$\begin{array}{ccc} \frac{1}{2\beta_{1}}\frac{d}{dt}\beta_{1}\;\;\; & = & \;\;\;-\frac{2\gamma }{\nu -1}\frac{I_{\nu }}{Z_{2}^{\nu }\sqrt{\beta_{2}}}\beta_{1}Z_{1}^{1-\gamma }\;\;\;,\;\;\;-\frac{1}{Z_{1}}\frac{d}{dt}Z_{1}=-\frac{2\gamma }{\nu -1}\frac{I_{\nu }}{Z_{2}^{\nu }\sqrt{\beta_{2}}}\beta_{1}Z_{1}^{1-\gamma }, \end{array}$$(45)$$\begin{array}{ccc} \frac{1}{2\beta_{2}}\frac{d}{dt}\beta_{2}\;\;\; & = & \;\;\;-\frac{2\nu }{\gamma -1}\frac{I_{\gamma }}{Z_{1}^{\gamma }\sqrt{\beta_{1}}}\beta_{2}Z_{2}^{1-\nu }\;\;\;,\;\;\;-\frac{1}{Z_{2}}\frac{d}{dt}Z_{2}=-\frac{2\nu }{\gamma -1}\frac{I_{\gamma }}{Z_{1}^{\gamma }\sqrt{\beta_{1}}}\beta_{2}Z_{2}^{1-\nu }, \end{array}$$
with
(46)$$\begin{array}{c} I_{\kappa }=\left\{\begin{array}{cc} \frac{\Gamma \left(\frac{1}{2}\right)\Gamma \left(1+\frac{\kappa }{\kappa -1}\right)}{\sqrt{\kappa -1}\Gamma \left(\frac{3}{2}+\frac{\kappa }{\kappa -1}\right)} & 1\leq \kappa <2\\ \frac{\Gamma \left(\frac{1}{2}\right)\Gamma \left(\frac{\kappa }{1-\kappa }-\frac{1}{2}\right)}{\sqrt{1-\kappa }\Gamma \left(\frac{\kappa }{1-\kappa }\right)} & 0\leq \kappa \leq 1 \end{array}\right.\;, \end{array}$$
where κ=γ or ν.

Figure 1 shows the behavior of the mean square displacement for two different sets of γ and ν in the absence of external forces. The values chosen for the parameters γ and ν are responsible for different behaviors of the mean square displacement for each case, as pointed out in the inset of Figure 1. In particular, the diffusion present in this scenario is anomalous [51,52]. Figure 2 shows the behavior of Equation (Equation 41) for two different sets of γ and ν. Note that different values of $\beta_{1}\left(0\right)$ and $\beta_{2}\left(0\right)$ used to obtain Figure 1 and Figure 2 are connected to different initial conditions for each subsystem. This is the reason why we initially verified different behaviors for each set of the parameters γ and ν, and, after some time, the mean square displacement has the same time dependence for both subsystems. The entropy production is shown in the inset in Figure 2, which corresponds to the behavior of Equation (Equation 61) for the entropy given by Equation (Equation 41). We underline that the system composed of these two systems reaches equilibrium in the limit of t→∞, since in this limit $\dot{S}\left(t\right)\to 0$. For general nonlinear Fokker–Planck equations, the entropy should simultaneously satisfy the following equations,
(47)$$\begin{array}{c} -\rho_{1}\frac{\partial^{2}}{\partial \rho_{1}^{2}}s\left(\rho_{1},\rho_{2}\right)=\frac{\partial }{\partial \rho_{1}}P_{1}\left(\rho_{1},t\right)\;\;\;\mathrm{and}\;\;\;-\rho_{2}\frac{\partial^{2}}{\partial \rho_{2}^{2}}s\left(\rho_{1},\rho_{2}\right)=\frac{\partial }{\partial \rho_{2}}P_{2}\left(\rho_{2},t\right), \end{array}$$
to verify $\frac{d}{dt}F\leq 0$ and, consequently, satisfy the H–theorem. It is also significant to mention that, depending on the form of the nonlinear dependence in the Equations (Equation 1) and (Equation 2), which may not recover the standard form of the Fokker–Planck equation, the entropy associated with these equations will not recover the usual form.

### 2.2. Entropy Production

Let us analyze the entropy production related to Equation (Equation 17) with the dynamics of $\rho_{1}\left(x_{1},t\right)$ and $\rho_{2}\left(x_{2},t\right)$ given by Equations (Equation 19) and (Equation 20). By performing a time derivative of Equation (Equation 17), we obtain
(48)$$\begin{array}{ccc} \frac{d}{dt}S\left(t\right) & = & k\int_{-\infty }^{\infty }\int_{-\infty }^{\infty }dx_{1}dx_{2}\left[\frac{\partial }{\partial \rho_{12}}s\left(\rho_{12}\right)\right]\frac{\partial }{\partial t}\left[\rho_{1}\left(x_{1},t\right)\rho_{2}\left(x_{2},t\right)\right]\\ & = & -k\int_{-\infty }^{\infty }dx_{1}\int_{-\infty }^{\infty }dx_{2}\rho_{2}\frac{\partial }{\partial \rho_{12}}s\left(\rho_{12}\right)\frac{\partial }{\partial x_{1}}J_{1}\left(x_{1},t\right)\\ & - & k\int_{-\infty }^{\infty }dx_{1}\rho_{1}\int_{-\infty }^{\infty }dx_{2}\frac{\partial }{\partial \rho_{12}}s\left(\rho_{12}\right)\frac{\partial }{\partial x_{2}}J_{2}\left(x_{2},t\right) \end{array}$$
and, consequently, performing integration by parts with the conditions $J_{1}\left(x_{1}\to \pm \infty ,t\right)\to 0$ and $J_{2}\left(x_{2}\to \pm \infty ,t\right)\to 0$, also
(49)$$\begin{array}{ccc} \frac{d}{dt}S\left(t\right) & = & k\int_{-\infty }^{\infty }dx_{1}\int_{-\infty }^{\infty }dx_{2}\left[\rho_{2}^{2}\frac{\partial^{2}}{\partial \rho_{12}^{2}}s\left(\rho_{12}\right)\frac{\partial \rho_{1}}{\partial x_{1}}\right]J_{1}\left(x_{1},t\right)\\ & + & k\int_{-\infty }^{\infty }dx_{1}\int_{-\infty }^{\infty }dx_{2}\left[\rho_{1}^{2}\frac{\partial^{2}}{\partial \rho_{12}^{2}}s\left(\rho_{12}\right)\frac{\partial \rho_{2}}{\partial x_{2}}\right]J_{2}\left(x_{2},t\right)\;. \end{array}$$
It is possible to simplify Equation (Equation 48) by using, from the H–theorem, the equations
(50)$$\begin{array}{c} -kT\rho_{1}\rho_{2}^{2}\frac{\partial \rho_{1}}{\partial x_{1}}\frac{\partial^{2}}{\partial \rho_{12}^{2}}s\left(\rho_{12}\right)=\Gamma \frac{\partial }{\partial x_{1}}P_{1}\left(\rho_{1},t\right) \end{array}$$
and
(51)$$\begin{array}{c} -kT\rho_{2}\rho_{1}^{2}\frac{\partial \rho_{2}}{\partial x_{2}}\frac{\partial^{2}}{\partial \rho_{12}^{2}}s\left(\rho_{12}\right)=\Gamma \frac{\partial }{\partial x_{2}}P_{2}\left(\rho_{2},t\right), \end{array}$$
in order to obtain
(52)$$\begin{array}{ccc} \frac{d}{dt}S\left(t\right) & = & -\frac{1}{T}\int_{-\infty }^{\infty }dx_{1}F_{1}\left(x_{1}\right)J_{1}\left(x_{1},t\right)-\frac{1}{T}\int_{-\infty }^{\infty }dx_{2}F_{2}\left(x_{2}\right)J_{2}\left(x_{2},t\right)\\ & + & \frac{1}{T}\int_{-\infty }^{\infty }dx_{1}\frac{J_{1}^{2}\left(x_{1},t\right)}{\rho_{1}\left(x_{1},t\right)}+\frac{1}{T}\int_{-\infty }^{\infty }dx_{2}\frac{J_{2}^{2}\left(x_{2},t\right)}{\rho_{2}\left(x_{2},t\right)}, \end{array}$$
where
(53)$$\begin{array}{c} J_{1}\left(x_{1},t\right)=-\Gamma \frac{\partial }{\partial x_{1}}P_{1}\left(\rho_{1},t\right)+F_{1}\left(x_{1}\right)\rho_{1}\left(x_{1},t\right) \end{array}$$
and
(54)$$\begin{array}{c} J_{2}\left(x_{2},t\right)=-\Gamma \frac{\partial }{\partial x_{2}}P_{2}\left(\rho_{2},t\right)+F_{2}\left(x_{2}\right)\rho_{2}\left(x_{2},t\right), \end{array}$$
with $P_{1}\left(\rho_{1},t\right)$ and $P_{2}\left(\rho_{2},t\right)$ given by Equations (Equation 10) and (Equation 11). Equation (Equation 48) can be written as follows:(55)$$\begin{array}{c} \frac{d}{dt}S=\Pi -\Phi \end{array}$$
where one identifies the entropy flux, representing the exchanges of entropy between the subsystems represented by $\rho_{1}$ and $\rho_{2}$ and their neighborhood,
(56)$$\begin{array}{c} \Phi =\frac{1}{T}\int_{-\infty }^{\infty }dx_{1}F_{1}\left(x_{1}\right)J_{1}\left(x_{1},t\right)+\frac{1}{T}\int_{-\infty }^{\infty }dx_{2}F_{2}\left(x_{2}\right)J_{2}\left(x_{2},t\right), \end{array}$$
as well as the entropy-production contribution:(57)$$\begin{array}{c} \Pi =\frac{1}{T}\int_{-\infty }^{\infty }dx_{1}\frac{J_{1}^{2}\left(x_{1},t\right)}{\rho_{1}\left(x_{1},t\right)}+\frac{1}{T}\int_{-\infty }^{\infty }dx_{2}\frac{J_{2}^{2}\left(x_{2},t\right)}{\rho_{2}\left(x_{2},t\right)}. \end{array}$$
We underline that *T* and $\rho_{i}\left(x_{i},t\right)$ are positive quantities, yielding the desirable result: Π≥0.

For the general case represented by Equation (Equation 4), we have
(58)$$\begin{array}{ccc} \frac{d}{dt}S\left(t\right) & = & k\int_{-\infty }^{\infty }\int_{-\infty }^{\infty }dx_{1}dx_{2}\left[\frac{\partial }{\partial \rho_{1}}s\left(\rho_{1},\rho_{2}\right)\frac{\partial }{\partial t}\rho_{1}\left(x_{1},t\right)+\frac{\partial }{\partial \rho_{2}}s\left(\rho_{1},\rho_{2}\right)\frac{\partial }{\partial t}\rho_{2}\left(x_{2},t\right)\right]\\ & = & -k\int_{-\infty }^{\infty }\int_{-\infty }^{\infty }dx_{1}dx_{2}\frac{\partial }{\partial \rho_{1}}s\left(\rho_{1},\rho_{2}\right)\frac{\partial }{\partial x_{1}}J_{1}\left(x_{1},t\right)\\ & - & k\int_{-\infty }^{\infty }\int_{-\infty }^{\infty }dx_{1}dx_{2}\frac{\partial }{\partial \rho_{2}}s\left(\rho_{1},\rho_{2}\right)\frac{\partial }{\partial x_{2}}J_{2}\left(x_{2},t\right). \end{array}$$
Performing integration by parts in Equation (Equation 58) and by taking into account the conditions $J_{1}\left(x_{1}\to \pm \infty ,t\right)\to 0$ and $J_{2}\left(x_{2}\to \pm \infty ,t\right)\to 0$, we obtain that
(59)$$\begin{array}{ccc} \frac{d}{dt}S\left(t\right) & = & k\int_{-\infty }^{\infty }\int_{-\infty }^{\infty }dx_{1}dx_{2}\left[\frac{\partial \rho_{1}}{\partial x_{1}}\frac{\partial^{2}}{\partial \rho_{1}^{2}}s\left(\rho_{1},\rho_{2}\right)\right]J_{1}\left(x_{1},t\right)\\ & + & k\int_{-\infty }^{\infty }\int_{-\infty }^{\infty }dx_{1}dx_{2}\left[\frac{\partial \rho_{2}}{\partial x_{2}}\frac{\partial^{2}}{\partial \rho_{2}^{2}}s\left(\rho_{1},\rho_{2}\right)\right]J_{2}\left(x_{2},t\right). \end{array}$$
By using the equations,
(60)$$\begin{array}{c} -\rho_{1}\frac{\partial \rho_{1}}{\partial x_{1}}\frac{\partial^{2}}{\partial \rho_{1}^{2}}s\left(\rho_{1},\rho_{2}\right)=\frac{\partial }{\partial x_{1}}P_{1}\left(\rho_{1},t\right)\;\;\;\mathrm{and}\;\;\;-\rho_{2}\frac{\partial \rho_{2}}{\partial x_{2}}\frac{\partial^{2}}{\partial \rho_{2}^{2}}s\left(\rho_{1},\rho_{2}\right)=\frac{\partial }{\partial x_{2}}P_{2}\left(\rho_{2},t\right), \end{array}$$
it is possible to simplify Equation (Equation 59) in order to obtain
(61)$$\begin{array}{ccc} \frac{d}{dt}S\left(t\right) & = & -\frac{1}{T}\int_{-\infty }^{\infty }dx_{1}F_{1}\left(x_{1}\right)J_{1}\left(x_{1},t\right)-\frac{1}{T}\int_{-\infty }^{\infty }dx_{2}F_{2}\left(x_{2}\right)J_{2}\left(x_{2},t\right)\\ & + & \frac{1}{T}\int_{-\infty }^{\infty }dx_{1}\frac{J_{1}^{2}\left(x_{1},t\right)}{\rho_{1}\left(x_{1},t\right)}+\frac{1}{T}\int_{-\infty }^{\infty }dx_{2}\frac{J_{2}^{2}\left(x_{2},t\right)}{\rho_{2}\left(x_{2},t\right)}, \end{array}$$
where
(62)$$\begin{array}{c} J_{1}\left(x_{1},t\right)=-\Gamma \frac{\partial }{\partial x_{1}}P_{1}\left(\rho_{1},t\right)+F_{1}\left(x_{1}\right)\rho_{1}\left(x_{1},t\right) \end{array}$$
and
(63)$$\begin{array}{c} J_{2}\left(x_{2},t\right)=-\Gamma \frac{\partial }{\partial x_{2}}P_{2}\left(\rho_{2},t\right)+F_{2}\left(x_{2}\right)\rho_{2}\left(x_{2},t\right), \end{array}$$
as before, with $P_{1}\left(\rho_{1},t\right)$ and $P_{2}\left(\rho_{2},t\right)$ arbitrary. Note that Equation (Equation 61) is formally equal to Equation (Equation 52), which evidences that the result obtained for the entropy production is invariant in form when the entropies are obtained from the H–theorem.

## 3. Discussion and Conclusions

We have investigated the entropy of a system composed of two subsystems governed by nonlinear Fokker–Planck equations. In this context, we have essentially analyzed two scenarios; in one of them, the subsystems have the same dynamics, and in the other one, they have different dynamics, i.e., the nonlinear Fokker–Planck equations are different. The first case allows the definition of entropy which can be connected to different cases and preserves the formal structure $S\left(\rho_{1},\rho_{2}\right)=S\left(\rho_{1}\rho_{2}\right)$ also verified by the standard entropy of Boltzmann–Gibbs. For the other case, we consider different dynamics for each subsystem, which allows the definition of an entropic form for which $S\left(\rho_{1},\rho_{2}\right)\neq S\left(\rho_{1}\rho_{2}\right)$. In both cases, we have analyzed the entropy production and we have shown the effect of each subsystem on the composite system. In addition, we have shown that the time variation of the entropy (entropy production) for the total system is invariant in form for all the cases considered here.

## Figures and Tables

**Figure 1 entropy-25-01357-f001:**
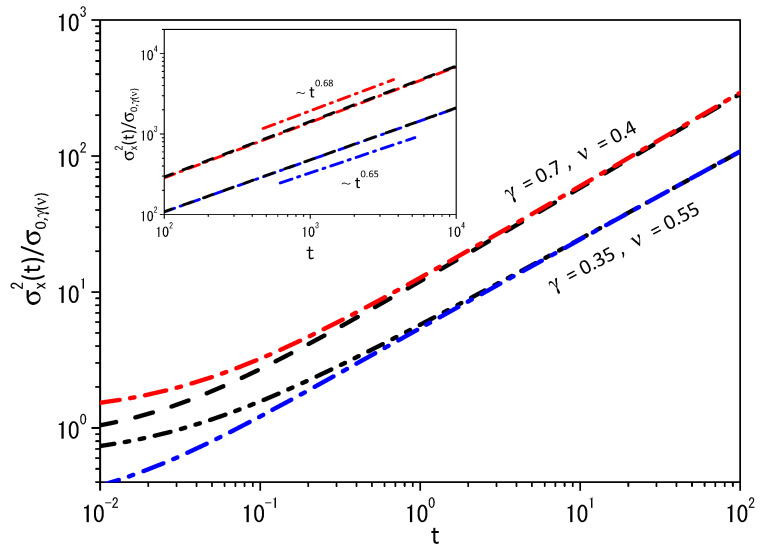
Behavior of $\sigma_{x}^{2}/\sigma_{0,\gamma (\nu )}$ versus *t* for two different sets of γ and ν, where $\sigma_{0,\gamma (\nu )}=\sigma_{\gamma (\nu )}\int_{-\infty }^{\infty }d\xi \xi^{2}\exp_{\gamma (\nu )}\left(-\xi^{2}\right)/\int_{-\infty }^{\infty }d\xi \exp_{\gamma (\nu )}\left(-\xi^{2}\right)$, where $\sigma_{\gamma (\nu )}$ is chosen in order to collapse the curves for each set of values. We consider, for simplicity, $\beta_{1}\left(0\right)=2$ and $\beta_{2}\left(0\right)=1$. The red dashed-dotted and black dashed lines represent the case γ=0.4 with ν=0.7. The blue dashed-dotted and black dashed-dotted-dotted lines represent the case γ=0.35 with ν=0.55. Notice that the behavior for the cases worked out in this figure have different time dependence for the mean square displacement, as pointed out in the inset.

**Figure 2 entropy-25-01357-f002:**
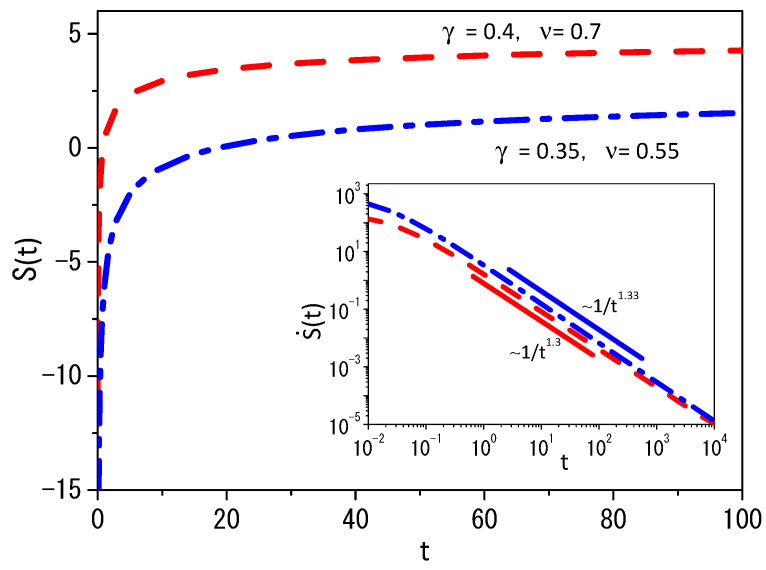
Behavior of Equation (Equation 41) versus *t* for two different sets of γ and ν. We consider, for simplicity, $\beta_{1}\left(0\right)=2$ and $\beta_{2}\left(0\right)=1$. The red dashed-dotted line represents the case γ=0.4 with ν=0.7. The blue dashed-dotted line represents the case γ=0.35 with ν=0.55. Notice that the behavior for the cases worked out in this figure have different time dependence for $\dot{S}\left(t\right)$, as pointed out in the inset.

## Data Availability

Not applicable.

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
