# Peer review of "Nonlinear Fokker–Planck Equations, H-Theorem and Generalized Entropy of a Composed System"

_entropy, 2023, doi:10.3390/e25091357_

Round 1
Reviewer 1 Report
The authors investigate the entropy of a system composed of two subsystems governed by nonlinear Fokker-Planck equations. Their calculations are worth the effort and bring progress to the field. Therefore, I recommend the article for publication.
Author Response
Please, see the attached file.

Reviewer 2 Report
The authors have investigated the entropy of a system consisting of two subsystems ruled by nonlinear Fokker–Planck equations (FPE).
They havve analyzed two distinct scenarios; 1) the subsystems have the same dynamics, and
2) they have different dynamics so that the nonlinear FPE are different.
The case 1) permits constructing an entropy that can be linked to distinct instnces preserving the formal relation S(ρ1, ρ2) = S(ρ1ρ2), The equality is verified as well by the standard Boltzmann-Gibbs entropy .
For the second case, the authors assign distinct dynamics tor each subsystem.
This permits advancing an entropic form for which S(ρ1, ρ2) = S(ρ1ρ2). In both
instances, the authors discuss the entropy production and display the effects that each subsystem producs on the composite system.
The authors also show that the time variation of the entropy production
of the total system remains form-invariant in all the cases looke at in this paper.
The paper is well written, relevant, and interesting.
Acceptance is to be recommended
Author Response
Please, see the attached file.

Reviewer 3 Report
This manuscript considers the entropy of a system composed of two subsystems governed by nonlinear Fokker–Planck equations (NFPEs).
The two NFPEs are independent in equation, that is, the NFPE for one system does not contain variables of the other. These are represented by Eqs. (1) and (2). Thus, the subsequent integration can be
performed independently. The coupling effect comes from the joint probability through the entropy and the internal energy in the free energy. The authors show that using the decreasing property of the
free energy in time, a particular form of the two-parameter generalized entropy is concluded under some assumptions for the diffusion constants.
The content might be relevant, but the presentation is hard to read. That is, some descriptions are not thorough. I list some concerns below.
The title of the manuscript is too general. It should reflect more specific content presented. In Eqs. (1) and (2), Gamma should be denoted. The authors use many undefined symbols,
for example, in Eq. (6), $\rho_{12}$ $\rho_{1,2}$ are not defined. Also, P without the subscripts 1 and 2 is not defined.
In addition, the formulas contain many typos:
In the third line of Eq. (7) (also in Eq. (34)), $\Phi(x_1,x_2)$ is wrong.
In Eq. (9), for the second term, $\rho_{1(2)}$ must be used.
In Eq. (12), the second and fourth lines should be divided by $\rho_2$ and $\rho_1$, respectively. Thus, Eq. (14) should be corrected.
These prevent the readers from understanding.
The authors must carefully check the calculations.
The partition functions for Eqs. (23) and (24) should contain $D_{1(2),\gamma}$.
The authors also consider the following concerns.
- It is unclear how Eq. (15) is derived from Eq. (14).
- Eq. (38) needs explanations why these hold from Eq. (37).
- The sentence in Page 7 Lines 148-149 is unclear.
- The readers will need the derivations for Eqs. (44) and (45).
- The expression $gamma(\nu)$ in Eq. (46) gives a misunderstanding of the functional relation.
- In Fig. 1, it is unclear what several different lines and colors denote.
- Page 8 line 161, what do you mean by 'the same time dependence'?
Do you mean that both slopes in Fig. 1 get pararell?
- Why does the general case imply Eq. (47)? It needs an explanation.
- Entropy is defined in Eq. (4). Then, why does Eq. (48) (also (55)) need the minus sign?
- Readers will need the derivations for Eq. (49).
The authors do not provide some relevant works on nonlinear FP equations. The authors should mention a standard reference book in the literature in Introduction.
The English errors must be corrected.
Author Response
Please, see the attached file.

Round 2
Reviewer 3 Report
The revised manuscript became better for readers.
It still remains concerns:
Page 2, zero's law -> zeroth law
Page 3 Eq. (10), the argument x -> x_i
Some arrors should be corrected.
Author Response
Dear Editor,
We thank the Referee for the very useful comments. We took advantage of this second round of revision to correct small typing errors (especially those pointed out in the indexes of the variables in the mathematical expressions, of which there were several). We appreciate the opportunity to reread the work and improve its presentation.
Thank you again for your support.
All the best,
Luiz R. Evangelista